# Removal of Zn^2+^ from Aqueous Solution Using Biomass Ash and Its Modified Product as Biosorbent

**DOI:** 10.3390/ijerph19159006

**Published:** 2022-07-24

**Authors:** Lei Xu, Xiangyu Xing, Jianbiao Peng

**Affiliations:** 1Henan Province Engineering Research Center of Environmental Laser Remote Sensing Technology and Application, Nanyang Normal University, Nanyang 473001, China; 2Collaborative Innovation Center of Water Security for Water Source Region of Mid-Line of South-to-North Diversion Project of Henan Province, Nanyang Normal University, Nanyang 473001, China; 3Non-Major Foreign Language Teaching Department, Nanyang Normal University, Nanyang 473061, China; xiangyuxing0902@163.com; 4School of Environment, Henan Normal University, Xinxiang 453007, China; pjb126com@163.com

**Keywords:** biomass ash, mesoporous structure, Zn^2+^, adsorption characteristics, productive reuse

## Abstract

To study the removal effect of bottom ash of biomass power plants and its modified products on zinc (Zn^2+^) in aqueous solution, a series of indoor experiments is carried out. The aim of this work is to explore a method to improve the ability of biomass ash to remove Zn^2+^ from aqueous solution and obtain its adsorption characteristics of Zn^2+^ in aqueous solution; on this basis, the feasibility of its application in the treatment of Zn^2+^-contaminated wastewater is analyzed. The mesoporous siliceous material is used to modify the biomass, and the modified material is functionalized with 3-aminopropyltriethoxysilane. The results show that the specific surface area of modified biomass ash is nine times that of the material before modification. The adsorption capacity of Zn^2+^ on the material increases with the increase of pH, and pH 6 is the optimum pH to remove Zn^2+^ from the aqueous solution. The Langmuir model and Freundlich model can show better fits for biomass ash and the modified material, respectively. Thermodynamic analysis results show that the adsorption of Zn^2+^ is spontaneous and endothermic in nature. The adsorption of Zn^2+^ onto biomass and modified biomass ash follow pseudo-first-order and pseudo-second-order kinetics, respectively.

## 1. Introduction

Heavy metal pollution in water has become a common global problem. Water bodies polluted by heavy metals have the following three characteristics: (1) heavy metals can be enriched in organisms, participate in the biological cycle, and accumulate in the biological chain through various channels, which lead to the accumulation in humans [1]. (2) Heavy metal pollutants are not easy to degrade and so exist permanently in the environment [2]. (3) Heavy metals also have strong toxicity under low concentration conditions and will even be transformed into other valence states with stronger toxicity under the action of microorganisms, which is a threat to the biosphere. The concentration range of its toxicity is generally 1.0–10.0 mg/L; even if it is below 1.0 mg/L, it will still affect the ecosystem [3]. Metal pollutants in the environment mainly come from anthropogenic industrial and agricultural activities, and they can enter the water system in different ways, such as atmospheric sedimentation, waste water irrigation, and slag leaching [4]. Zinc is one of the most common and widely distributed heavy metals in the environment, and as an essential element for many organisms, it is beneficial to organisms when the content does not exceed the standard. However, due to industrial activities such as smelting, electroplating, mining, plastic manufacturing, and metallurgy, a large amount of wastewater carrying Zn^2+^ is discharged into the environment; moreover, Zn^2+^ is not easy to degrade in the environment, resulting in the content of zinc in water bodies increasing [5]. A large amount of Zn^2+^ accumulation will cause a series of negative effects on the health of organisms, mainly manifested as neurological symptoms, and can even lead to brain tissue atrophy. Therefore, it is of essential significance to reduce the concentration of Zn^2+^ in industrial wastewater by technical means before it is discharged into the environment, so as to reduce its impact on the environment.

The traditional methods of removing heavy metals from aqueous solutions include chemical precipitation, solvent extraction, ion exchange, membrane separation, and electrolysis [6]. However, most of these methods are uneconomical, such as by consuming a lot of energy [7,8]. In addition, when the concentration of heavy metals in water is low, these methods have the problems of low efficiency or high cost. Moreover, they may produce secondary waste that is more difficult to treat than raw wastewater [9,10]. These shortcomings have prompted researchers to seek both economical and efficient technologies to treat heavy metal-contaminated water [11,12]. Recently, the use of environmentally friendly materials to remove heavy metals from a large amount of wastewater has aroused the interest of researchers. Agricultural wastes and by-products are widely regarded as cheap adsorbents for removing toxic metals from solutions. In the past decade, these materials have been widely used as adsorbents to replace existing technologies. For example, *Melaleuca diosmifolia* leaf [13], tomato leaf powder [14], rice husk [15], pine cone [16], olive pomace [17], pineapple stem [18], coffee husk [19], coffee waste [20], cauliflower leaf [21], rubber leaf [22], Formosa papaya seeds [23], parsley stalks [24], potato peel waste [25], cucumber peel [26], and water hyacinth [27] have been studied, and it has been found that these agricultural wastes have fine remediation effects on metal polluted wastewater. Compared with other adsorbents, these adsorbents not only have the advantages of fine remediation effects but also are agricultural wastes, which can be widely obtained and are inexpensive. Through this remediation mode, not only can the problem of water contamination be solved, but the problem of agricultural environmental pollution can also be solved.

Recently, the ash of agricultural wastes and by-products have been reported to have good adsorption properties for heavy metal ions in wastewater [28,29]. The results show that biomass ash can remove heavy metal ions from wastewater, mainly due to the high proportion of unburned C and Si present in these materials [30,31]. At the same time, after agricultural waste or by-products are burned, most of this ash has a large specific surface area. These characteristics give them adsorption properties, which are conducive to the adsorption of heavy metals in aqueous solution. However, the potential mechanism of metal removal from aqueous solution by this adsorbent is not fully understood.

According to the current situation of biomass power plants in China, most power plants use crop straw as fuel, which produces a large amount of biomass ash in the process of power generation. As an industrial waste, biomass ash cannot be used and abandoned, which leads to the accumulation of a large amount of biomass ash in the environment [32]. Considering the good adsorption capacity of to heavy metal ions in aqueous solution, research has shown that the maximum adsorption capacity for Pb^2+^, Ni^2+^, Cd^2+^, Mn^2+^, Zn^2+^, and Cr^3+^ in aqueous solution reached 1.95, 2.23, 2.00, 2.49, 2.46, and 1.50 g/kg [33], respectively; biomass ash can be used as an economic and environmentally friendly adsorbent. However, it has been reported that the adsorption capacity of natural biomass ash for specific heavy metals in aqueous solution is lower than that of some commercial or modified adsorbents [34]. This leads to the low efficiency of using natural biomass ash collected from biomass power plants as an adsorbent to remove heavy metal ions from industrial wastewater. The adsorption capacity of biomass ash to heavy metals in wastewater can be improved through appropriate modification methods so as to improve its ability to remediate heavy metal-polluted wastewater [35]. Therefore, it is necessary to explore a suitable method for modifying biomass ash.

Various mesoporous materials based on silica have been widely studied and partially commercialized. These modified materials have good adsorption properties for heavy metal ions in aqueous solution because of their large specific surface area (2–50 nm), high thermal stability, good mechanical stability, uniform pore morphology, and high functionality [36]. In addition, by combining specific organic functional groups on the surface and/or in the pores of mesoporous materials, the removal rate of mesoporous materials for Zn^2+^, Cu^2+^, Pb^2+^, and Cr^3+^ have been improved by 15.2–36.8%, 4.7–43.6%, 29.8–41.1%, and 20.5–39.4%, respectively, which gives them better application prospects [37,38,39]. Some researchers have synthesized a new material from coal fly ash and functional mesoporous materials; this new material has good adsorption properties for various pollutants in aqueous solution, including heavy metal ions [40]. In fact, compared with coal fly ash, biomass ash has a high silicon content, which makes it also possible to become a silica skeleton so as to carry out mesoporous modification and improve its adsorption performance. However, no research on mesoporous modification of biomass ash has been reported.

Therefore, the purpose of this study is to modify biomass ash with mesoporous silica and organosilane so as to improve its ability to remove Zn^2+^ from aqueous solution. On this basis, we obtained their adsorption characteristics of Zn^2+^ in aqueous solution, and the feasibility of its application in the treatment of Zn^2+^-contaminated wastewater was analyzed. The research results will help to understand the removal effect and conditions of this new material so as to provide a theoretical basis for their industrial promotion to remove Zn^2+^ from wastewater.

## 2. Materials and Methods

### 2.1. Biomass Ash

The biomass ash sample used in this study was taken from a biomass power plant burning agricultural residues in Anhui Province, China. The power plant uses a mixture of wheat straw, corn straw, peanut shell, and cotton straw as fuel for power generation. This mixture of fuels is burned in a mobile grate furnace at an excess air temperature of 850 °C.

### 2.2. Modification Experiment

The biomass ash was modified by co-condensation in a hexagonal mesoporous silica (HMS) matrix [41], and the synthesis steps reported by Walcarius et al. [42] were used. First, 1.24 g of dodecylamine was dissolved in 10 mL of alcohol followed by the addition of a mixture of 1.24 g of biomass ash in 90 mL of ultrapure water (CN61 M-UPR-I-20L) under stirring at 1000 rpm. Next, 6.09 mL of tetraethyl orthosilicate and 0.71 mL of 10% (*w/v*) 3-aminopropyltriethoxysilane [APS, NH_2_(CH_2_)_3_Si (OC_2_H_5_)_3_], an organosilane, were added into the reaction mixture. After 30 s, 0.94 mL of trimethylbenzene was added, and the mixture was then stirred for 24 h. Finally, the mixture was filtered through a 0.45 µm filter membrane, and the residue was air dried at room temperature. The remaining trimethylbenzene was Soxhlet extracted with 125 mL of alcohol for 5 h, and the sample was air dried at room temperature for 24 h.

### 2.3. Physico-Chemical Characterization and Surface Properties

In this paper, the elemental composition of biomass ash was determined by ICP-OES (Perkin Elmer optima 2000, Agilent Technologies Inc., Santa Clara, CA, USA), and a scanning electron microscope (SEM, semitachi s-4800, Hitachi, Tokyo, Japan) was used to determine the surface morphology. The functional group composition of biomass ash was determined by Fourier transform infrared spectroscopy (FTIR, Spectrum Two IR Spectrometers, Perkin Elmer, Shimadzu, Kyoto, Japan) at 4000–5000 cm^−1^, and the specific surface area of the sample was calculated by the BET method. The pH was measured using a METTLER TOLEDO pH meter (S40 SevenMulti^TM^, Mettler Toledo, Columbus, OH, USA) (solid/liquid ratio 1:5) [43].

### 2.4. Adsorption Experiments

The adsorption characteristics of the materials were assessed by evaluating the initial Zn^2+^ concentration, pH, and kinetic and thermodynamic factors. A total of 1000 mg/L Zn(NO_3_)_2_ standard solution was used to prepare the zinc solution used in the experiment. Then 0.1 M HNO_3_ and 0.1 M NaOH were used to adjust the solution pH. To ensure the reliability of the experimental results, the reagents used in the experiment were all G.R. (guaranteed reagent). The zinc adsorption capacity in the adsorption experiments was evaluated by Formula (1).
*A*_*Zn*_^2+^ = (*C_e_* − *C*_0_)*V*/*M*(1)
where *A_Zn_^2+^* is the zinc adsorption capacity, *C_e_* is the solution concentration after adsorption (mg/L), *C*_0_ is the solution concentration before adsorption (mg/L), *V* is the volume of the solution (L), and *M* is the mass of the adsorption material (g).

#### 2.4.1. Effect of pH

The pH greatly influences the removal of heavy metals in aqueous solution. To obtain the optimized pH value of biomass ash and modified materials for the removal of Zn^2+^, 0.1 g biomass and the modified material were separately added to 50 mL centrifuge tubes, and then 25 mL aliquots of solution (Zn^2+^ concentration: 50 mg/L; pH: 2.0–8.0) were separately added to the centrifuge tubes [34]. The centrifuge tubes were placed in a constant temperature shaker for 24 h (150 rpm, 25 °C).

#### 2.4.2. Adsorption Equilibrium Experiment

Solutions containing Zn^2+^ with concentrations of 50, 60, 70, 80, 90, and 100 mg/L were prepared by Zn^2+^ standard solution, and the pH value was adjusted to 5 with pH regulating solution. A total of 0.1 g of biomass ash and modified material was accurately weighed into a 50 mL centrifuge tube, and 25 mL amounts of the Zn^2+^ solutions with different concentrations listed above were and placed in a constant temperature shaker for 24 h (25 °C, 150 rpm). Then the centrifuge tubes were taken out and placed in the centrifuge for 10 min (3000 rpm). After filtering the solution with 0.45 μm micron microporous filter membrane, the Zn^2+^ concentration was measured with flame atomic absorption spectrometry (SpectrAA-220, Varian, Palo Alto, CA, USA).

#### 2.4.3. Adsorption Kinetics

The adsorption kinetics of Zn^2+^ were determined by adding 0.2 g of material into centrifuge tubes containing 100 mL Zn^2+^ solution (100 mg/L, pH 5); all centrifuge tubes were placed in a reciprocating shaker and shaken at a speed of 150 rpm for 24 h at 0.5, 1, 2, 3, 5, 10, 15, 30, 60, 90, 120, 180, and 240 min. Then 5 mL aliquots of samples were collected using a pipette (Eppendorf, Research Plus, 0.5–5 mL). The determination of Zn^2+^ was the same as in the adsorption isotherm experiment.

#### 2.4.4. Thermodynamic Studies

For thermodynamic studies, 0.1 g material were separately added to 50 mL centrifuge tubes containing 25 mL solution (Zn^2+^ concentration: 50 mg/L; pH: 5.0) and shaken at 30 °C, 45 °C, and 60 °C.

### 2.5. Data Processing

Data processing and analysis of variance were performed using Microsoft Excel 2010 (Microsoft Corporation, Redmond, WA, USA) and SPSS 20.0 (IBM SPSS, summers, New York, NY, USA). Graphics were conducted by Sigmaplot (12.5, Systat, San Jose, CA, USA).

## 3. Results and Discussion

### 3.1. Physico-Chemical Characterization and Surface Properties

The elemental composition of the biomass ash was reported by our previous study, and Si (12.0%), Ca (4.31%), and K (3.31%) were the main constituent elements [44]. SEM analysis showed that biomass ash was mainly composed of spherical particles and flake particles, and the particle diameter was 10–60 μm [34]. At the same time, we found that these ash particles were fully dispersed. The characterization results of these two materials were significantly different. After modification, the specific surface area of biomass ash was significantly increased, and its surface became smoother [34]. In addition, some weak channels were observed in the modified biomass ash, and the generation of pores helped to improve the porosity of the modified biomass ash so as to increase its adsorption sites (Table 1).

FTIR analysis showed that the modified biomass ash demonstrated an intense absorption band at 3330.62 cm^−1^, which could be attributed to the appearance of an O–H bond of the silanol group in modified biomass ash [34]. At the same time, we found that there were obvious absorption peaks at 845.85 cm^−1^ and 1051.82 cm^−1^, which corresponded to symmetric and asymmetric Si–O–Si vibrations, respectively. Compared with biomass ash, the spectral characteristics of the material functionalized with 10% (*w/v*) APS and HMS matrix changed significantly, mainly showing a broad signal between 3000 cm^−1^ and 3600 cm^−1^, which might be due to the increase in the number of silanol groups in the synthetic material. The stretching bands could be due to the N–H group of APS, and the band at 1488.2 cm^−1^ might be attributed to the bending vibration of the N–H groups [34,43]. The results of element composition analysis showed that C, O, Si, Al, Fe, and K existed in the two materials [34].

### 3.2. Effect of pH

The Zn^2+^ adsorption was found to be highly pH-dependent; moreover, in the pH range of 2–8, the modified material far exceeded that of biomass ash in terms of adsorption capacity (Figure 1). When the pH in the system was less than 4, the H^+^ concentration in the solution was very high, and the adsorption capacity of biomass ash to Zn^2+^ was very low, which might be due to the competitive adsorption of Zn^2+^ and H^+^ at low pH [45,46,47]. When the solution pH was considerably low, the value of H_3_O^+^ exceeded that of the Zn^2+^, and most adsorption sites on the material surface were occupied by H_3_O^+^, thereby reducing the adsorption capacity for the metal ion [48]. As the pH gradually increased, the concentration of H_3_O^+^ decreased and was gradually removed from the material surface. Consequently, the competitiveness between Zn^2+^ and H_3_O^+^ was decreased, so that metal ions could approach the active adsorption site on the material so as to increase the binding between Zn^2+^ and the surface of the synthetic matrix through ion exchange, resulting in improving the adsorption capacity [49,50]. The exchange mechanism between H^+^ and Zn^2+^ in solution can be expressed by the following equations:XOH + H_3_O^+^ → XOH_2_ + H_2_O(2)
XOH + OH^−^ → XO^−^ + H_2_(3)
2(XO^−^) + M^2+^ → (XO)_2_M(4)

X: Si, Fe, and Al.

M: metal.

The maximum adsorption efficiency of biomass ash and modified materials for Zn^2+^ were found to be near pH 6. When pH was >6, the adsorption of Zn^2+^ was weak, which can be attributed to the precipitation of Zn^2+^ species such as carbonates or hydroxides (Figure 1) [51]. The modified material was functionalized with NH_2_ groups, so that the material formed an amino-Zn complex with a greater stability constant after adsorbing Zn^2+^ in the solution, and the stability of this complex mainly depended on the pH of the solution system, which must be close to 7 [52].

### 3.3. Adsorption Isotherm

We used Langmuir and Freundlich adsorption models to fit the adsorption process of the material. The parameters of Langmuir and Freundlich are listed as follows [53,54,55]:*C_e_*/*q_e_* = 1/*q_L_*·*K_L_* + *C_e_*/*q_L_*(5)
(6)$$\ln(q_{e})=\ln(K_{F})+\frac{1}{n}\ln(C_{e})$$
where *C_e_* represents the equilibrium concentration of the metal ions (mg/L), *q_e_* represents the amount of metal ions adsorbed by a unit mass adsorbent (mg/g), *q_L_* represents the maximum amount of the metal ions adsorbed by the unit mass adsorbent (mg/g), and *K_L_* represents the Langmuir constant (L/mg). *K_F_* and *n* are the Freundlich constants, which indicate the adsorption capacity and adsorption intensity of a given material, respectively.

Through analysis, we found that the fitting of biomass ash by the Langmuir model was more optimized, while the fitting of the modified products by the Freundlich model was more optimized (Table 2). This might be because the adsorption of Zn^2+^ by biomass ash belonged to monolayer adsorption, so the experimental data could be well simulated by the Langmuir model at all temperature levels. However, the adsorption of Zn^2+^ by modified biomass ash belonged to multilayer adsorption. In addition to adsorbing Zn^2+^ on the surface through physical action, the functional groups on the surface of the adsorbent also existed in the form of Schiff bases (–N=CH–), and the –N=CH– could complex with Zn^2+^ in the solution. This indicated that Zn^2+^ occurred in both the adsorption reaction and complexation reaction on the modified material surface. This might be the fundamental reason why the adsorption process of modified materials for Zn^2+^ in solution did not conform to the Langmuir model. In the Freundlich model, the constant $\frac{1}{n}$ represented the adsorption strength. When the value of $\frac{1}{n}$ was between 1 and 10, the adsorption process was favorable [56]. In our study, the value of $\frac{1}{n}$ at each temperature was more than three, indicating that the modified biomass ash had good adsorption for Zn^2+^ (Table 2). By analyzing the data, we found that the adsorption capacity of both materials increased slightly with the increase of temperature. This might be because the adsorption process of the material to Zn^2+^ in the solution was an endothermic reaction, and thus increasing the temperature could increase the internal structure of the material and improve the adsorption capacity [57].

### 3.4. Thermodynamic Studies

The thermodynamic process and the parameters can be expressed according to Gupta [58]:ΔG^0^ = −*RT*ln*K_L_′*(7)
(8)$$\ln\frac{K_{L2}}{K_{L1}}\;=-(\Delta H_{0}/R)(\frac{T1-T2}{T2T1})\;$$

(9)$$\Delta S_{0}=\frac{{\Delta H}^{0}-{\Delta G}^{0}}{T}$$
where *K_L_’*, *K_L_*_1_, and *K_L_*_2_ are the Langmuir constants at *T*, *T*1, and *T*2, respectively; *R* is the gas constant (8.314 J·mol^−1^·K^−1^).

According to thermodynamics, ΔG is the adsorption driving force, which reflects the intensity of the adsorption driving force and depends on the enthalpy and entropy factor. ΔG is negative, and ΔH and ΔS are positive, indicating that the main driving force in the adsorption process is entropy change (Table 3). At the same time, the negative value of ΔG indicated that Zn^2+^ tended to be adsorbed from the solution to the modified biomass ash. Or we could understand it by this way—that the adsorption of Zn^2+^ on the material was spontaneous. As the temperature increased, ΔG decreased gradually, which indicated that the increase of temperature was conducive to the adsorption process; this result was also consistent with the endothermic process of the material’s adsorption of Zn^2+^ in the solution. In this research, the composition of the synthetic matrix ΔH was 30.0, indicating that the adsorption force was hydrogen bonding, which was ligand exchange [59]. This once again verified that the adsorption process of modified biomass ash for Zn^2+^ in solution included physical adsorption and chemical adsorption.

### 3.5. Kinetic Adsorption Studies

The adsorption kinetics of Zn^2+^ on biomass ash and modified products in the solution are shown in Figure 2. The Zn^2+^ concentration in aqueous solution decreased sharply in the initial 30 min, and the concentration of Zn^2+^ in the solution decreased to nearly 10 mg/L when the experiment lasted for 120 min. However, it took a longer time to reach the adsorption equilibrium for biomass ash. At the same time, when the adsorption process was close to equilibrium, the equilibrium concentration of Zn^2+^ was about five times that of the modified biomass ash, and the equilibrium removal rate of modified biomass ash (85.0%) was much higher than that of biomass ash (63.5%). This showed that the adsorption capacity of the modified biomass ash was much higher than that of the original biomass ash. This was also consistent with the results obtained by the Langmuir and Freundlich model. In the initial stage of adsorption reaction, there were a large number of active adsorption sites on the material surface; these active adsorption sites could provide space for Zn^2+^, which made Zn^2+^ move quickly to the material surface and be adsorbed and fixed [60]. However, with the progress of the adsorption process, the active adsorption sites on the material surface gradually decreased; therefore, the adsorption rate decreased rapidly [61]. The slow diffusion of Zn^2+^ on the internal matrix of modified biomass ash might lead to the reduction of the adsorption rate at this stage [62]. In order to simulate the change of adsorption rate of these two materials, pseudo-first-order and pseudo-second-order rate equations were used. The two models and the parameters can be expressed according to Zahra [63]:ln(*Q_e_ − Q_t_*) = ln*Q_e_* − *k*_1_t(10)
(11)$$\frac{t}{Q_{t}}\;=\frac{1}{k_{2}Q_{e}{}^{2}}+\frac{t}{Q_{e}}$$
where *Q_e_* is the adsorption capacity (mg/g) at equilibrium, *Q**_t_* is the amount (mg/g) of material adsorbed at time *t*, *k*_1_ represents the rate constant (min^−1^) of the pseudo-first-order model, and *k*_2_ is the rate constant (g/mg/min) of the pseudo-second-order model. For biomass ash, the *R^2^* value simulated by the pseudo-first-order model was much greater than that simulated by the pseudo-second-order model; however, for the modified biomass ash, this result was just the opposite (Table 4). For the modified biomass ash, the *R^2^* simulated by the pseudo-second-order model reached 1.00, which showed that the pseudo-second-order model could accurately simulate the adsorption process of Zn^2+^ by the modified biomass ash. The results of kinetic model simulation was consistent with the fitting results of Langmuir and Freundlich. The chemical adsorption process of modified materials for Zn^2+^ in solution might be caused by the reaction force and coordination process between Zn^2+^ and –NH_2_, –NH on the surface of modified biomass ash.

## 4. Conclusions

Using biomass ash as raw material, the mesoporous material synthesized by modification has a stronger adsorption capacity for Zn^2+^ and a higher removal rate of Zn^2+^ in aqueous solution. The adsorption capacity of the material for Zn^2+^ is closely related to the initial Zn^2+^ concentration and pH of the aqueous solution. Compared with untreated biomass ash, due to the functionalization of mesoporous materials by APS, the specific surface area of modified biomass ash is nine times that of the material before modification, which makes this material have more active adsorption sites to adsorb Zn^2+^ from the solution. The adsorption of Zn^2+^ by biomass ash conforms to the Langmuir model, while the adsorption of Zn^2+^ by modified biomass ash conforms to the Freundlich model. This is mainly due to the difference of adsorption mechanisms between the two materials. The adsorption process of Zn^2+^ by the two materials is endothermic. The adsorption process of Zn^2+^ by biomass ash conforms to the pseudo-first-order kinetics model (*R*^2^ = 0.968), while the adsorption process of Zn^2+^ by the modified material conforms to the pseudo-second-order kinetics model (*R*^2^ = 1.00). It is worth noting that compared with the reported materials, this modified material shows strong adsorption capacity for Zn^2+^ (its removal rate of Zn^2+^ in solution is 21.6% higher than that of biomass ash) and has great potential in the remediation of Zn^2+^ pollution in a water environment. This study provides a suitable method for the resource utilization of by-products of biomass power plants. However, more research is needed on its industrialization.

## Figures and Tables

**Figure 1 ijerph-19-09006-f001:**
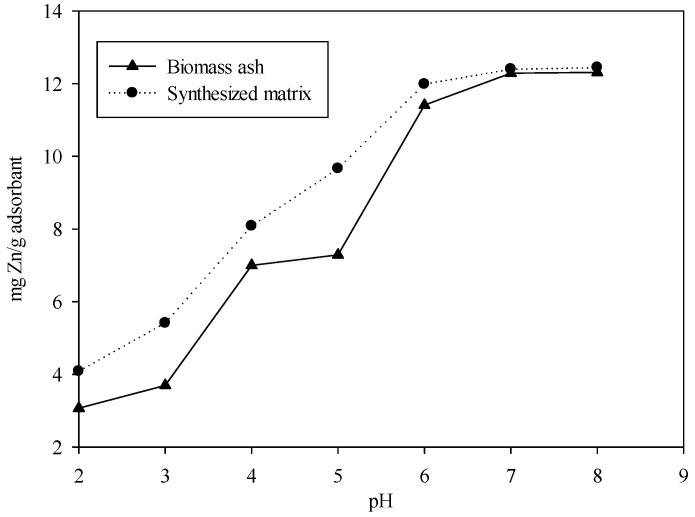
Effect of pH on the adsorption of Zn^2+^ on biomass ash and modified biomass ash (initial concentration of Zn^2+^, 50 mg/L; biomass ash concentration, 4 g/L; T = 30 °C).

**Figure 2 ijerph-19-09006-f002:**
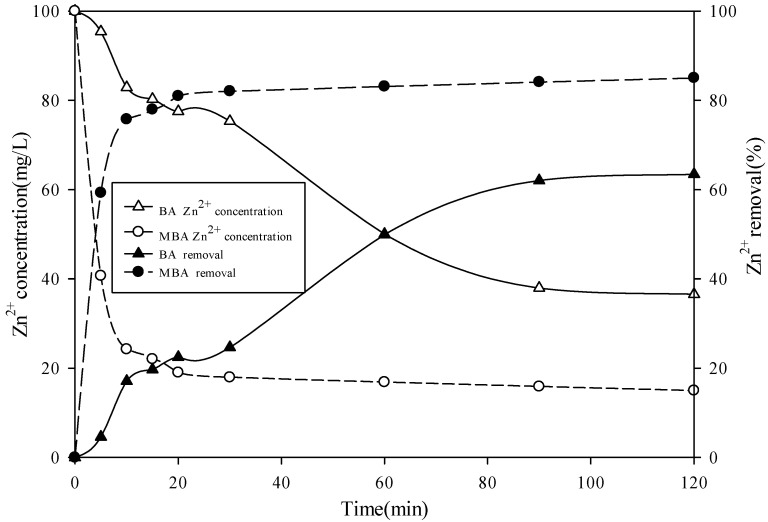
Sorption kinetics of the materials (initial concentrations of Zn^2+^, 100 mg/L; biomass ash and modified product concentration, 2 g/L; T = 30 °C; initial pH 5.0; BA = biomass ash, MBA = modified biomass ash).

**Table 1 ijerph-19-09006-t001:** Comparison of the Brunauer–Emmett–Teller (BET) analysis of functionalized hexagonal mesoporous silica, biomass ash, and synthesized matrix. Reproduced with permission from our previous research results [34]. HMS-NH2 is non-functionalized mesoporous silica.

Analysis	Sample		
	HMS-NH2 [44]	Biomass Ash	Modified Biomass
BET surface area (m^2^/g)	17	21.4 ± 0.17	186 ± 0.15

**Table 2 ijerph-19-09006-t002:** Values of the constants and fitting of the adjusted adsorption models.

Adsorbent	Temp (°C)	Langmuir			Freundlich		
		*q_L_* (mg/g)	*K_L_* (L/mg)	*R* ^2^	*n*	*K_F_* (mg/g) (mg/L)^1/*n*^	*R* ^2^
Biomass ash	30	20.0	1.19	1.00	5.65	11.9	0.931
	45	20.0	1.14	1.00	6.37	12.5	0.899
	60	20.4	1.15	0.999	6.41	12.8	0.915
Modified biomass ash	30	26.3	1.09	0.992	3.45	14.1	0.992
	45	27.0	1.23	0.993	3.36	14.3	0.999
	60	23.3	2.05	0.993	3.41	14.6	0.999

**Table 3 ijerph-19-09006-t003:** Thermodynamic parameters of biomass ash and the synthesized matrix.

	Temp (°C)	Thermodynamic Parameters		
		ΔG^0^ (kJ/mol)	ΔH^0^ (kJ/mol)	ΔS^0^ (J/mol/K)
Biomass ash	30	−0.438		
	45	−0.347	0.513	2.70
	60	−0.363		
Synthesized matrix	30	−0.217		
	45	−0.548	30.0	96.0
	60	−1.99		

**Table 4 ijerph-19-09006-t004:** Kinetic parameters for biomass ash and the synthesized matrix.

Adsorbent	Temp (°C)	Pseudo-First-Order Model		Pseudo-Second-Order Model	
		*k*_1_ (min^−1^)	*R* ^2^	*k*_2_ (g/mg/min)	*R* ^2^
Biomass ash	30	1.51 × 10^−2^	0.968	4.07 × 10^−4^	0.775
Synthesized matrix		1.12 × 10^−2^	0.445	2.91 × 10^−2^	1.00

## Data Availability

The data that support the findings of this study are available on request from the corresponding author.

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
