# Peer review of "Removal of Zn2+ from Aqueous Solution Using Biomass Ash and Its Modified Product as Biosorbent"

_ijerph, 2022, doi:10.3390/ijerph19159006_

Round 1

Reviewer 1 Report

The work describes an interesting study of the use of modified biomass for Zn removal from wastewater. The results of the carried investigation are well addressed and described. However, the English should be generally improved and both Abstract and Conclusions sections should be more quantitative and not so general.

1.       Line 31: “… various channels which lead to the accumulation in human[1].”. Please change human to “humans”.

2.       Line 48: “..as neurological symptoms, open eyes and coma.”. What do the authors intend to say with open eyes?

3.       Line 50: “Zn2+ chloride”. Please change to “zinc chloride” or “ZnCl2”.

4.       Line 62: “waste water”. Please change it to “wastewater”.

5.       Line 85: “can’t”. Please change it to “cannot”.

6.       Lines 87-90: The authors mentioned the good adsorption capacity of biomass and point out that some commercial adsorbents showed to have improved capacity when compared to biomass. However, to which cations are the authors referring to? It is also Zn2+? Can you provide some quantitative analysis from literature?

7.       Line 99: It is mentioned the large specific area of the mesoporous materials. But quantitatively, which values are the authors referring to? Which is the literature range for these materials.

8.       Line 103: How much these modifications improve the adsorption capacity?

9.       Line 133: “…by BET method, The pH was measured…”. There is a small typo here. Please remove the “,” and change it to “.”.

10.   In the experimental section, there is no reference to the labels of the reagents used and their grade.

11.   Lines 144-145: the volumes values of 50 and 25 mL are written in the text as “ml”. Please change it to “mL”.

12.   Line 149: Please change the beginning of the sentence to “Solutions containing Zn2+ were prepared…” instead of “Prepare Zn2+ solutions…”. Also change all verbs to the past tense.

13.   Line 149: “100 mg L-1”. In the manuscript the authors usually use mg/L but here mg L-1 is written instead. Please uniform it to always use the same style in the full manuscript. The same with all volumes mentioned since in the lines 151 and 152 “ml” is written instead of “mL”. Please check the rest of the manuscript regarding these issues.

14.   Line 171: “were conduct”. Please change it to “were conducted”.

15.   Line 173: “Physico-chemical characterization of and surface properties”. Please remove the “of” word.

16.   Line 174: “element composition”. Please change it to “elemental composition”.

17.   Line 175 and 193: Please write “elements” instead of “element”. Moreover, please write the wt% of each major element after their name, otherwise the readers will need to search for your previous publication to check the ICP results.

18.   Line 178: “these” instead of “this” because you are referring to the plural.

19.   Line 177: The full Fig.1 is equal to the previous study performed by the authors in Ref. [45]. The same image cannot be published two times in different papers and Journals. Since the previous study focused in Cu2+ with the same adsorbent, the authors should only mention in this manuscript that the SEM, FTIR and BET characterization were previously performed in [45], without copying the same tables and figures. The values, on the other hand, can be reported in the text but always mention that they were made in [45]. If the authors would like to introduce some extra characterization in this manuscript, I would suggest they report EDS spectra and/or EDS mapping images of the biomass.

20.   The experiment was performed how many times? The graphics in the manuscript do not have error bars. Is the given data reproducible?

21.   Line 267: “…verified that he adsorption…”. Typo in “he”, since it should be “the”.

22.   The conclusions are too general. Some quantification on the zinc abatement is required.

Author Response

Dear Editor,

We are very grateful for you and the reviewers’ comments and suggestions about this paper. Those comments are very valuable and helpful for revising and improving our manuscript. According to these revision suggestions, we have carefully addressed the comments and made revision in detail as following, and all the modified parts in the manuscript have been marked in red.

Sincerely yours,

Lei Xu.

Issues raised by Reviewer 1

Comments: The work describes an interesting study of the use of modified biomass for Zn removal from wastewater. The results of the carried investigation are well addressed and described. However, the English should be generally improved and both Abstract and Conclusions sections should be more quantitative and not so general.

  1. Line 31: “… various channels which lead to the accumulation in human[1].”. Please change human to “humans”.

Reply and revision: Thank you very much for your suggestion. We have revised it in the manuscript (L32).

  1. Line 48: “..as neurological symptoms, open eyes and coma.”. What do the authors intend to say with open eyes?

Reply and revision: Thank you very much for your suggestion. We have revised it in the manuscript (L49).

  1. Line 50: “Zn2+chloride”. Please change to “zinc chloride” or “ZnCl2”.

Reply and revision: Thank you very much for your suggestion. This sentence has been deleted in the manuscript.

  1. Line 62: “waste water”. Please change it to “wastewater”.

Reply and revision: Thank you very much for your suggestion. We have revised it in the manuscript (L60).

  1. Line 85: “can’t”. Please change it to “cannot”.

Reply and revision: Thank you very much for your suggestion. We have revised it in the manuscript (L84).

  1. Lines 87-90: The authors mentioned the good adsorption capacity of biomass and point out that some commercial adsorbents showed to have improved capacity when compared to biomass. However, to which cations are the authors referring to? It is also Zn2+? Can you provide some quantitative analysis from literature?

Reply and revision: Thank you very much for your suggestion. We have revised it in the manuscript (L87-89).

  1. Line 99: It is mentioned the large specific area of the mesoporous materials. But quantitatively, which values are the authors referring to? Which is the literature range for these materials.

Reply and revision: Thank you very much for your suggestion. We have revised it in the manuscript (L100).

  1. Line 103: How much these modifications improve the adsorption capacity?

Reply and revision: Thank you very much for your suggestion. We have revised it in the manuscript (L104-105).

  1. Line 133: “…by BET method, The pH was measured…”. There is a small typo here. Please remove the “,” and change it to “.”.

Reply and revision: Thank you very much for your suggestion. We have revised it in the manuscript (L144).

  1. In the experimental section, there is no reference to the labels of the reagents used and their grade.

Reply and revision: Thank you very much for your suggestion. We have added the grade of the reagents in the manuscript (L151-152).

  1. Lines 144-145: the volumes values of 50 and 25 mL are written in the text as “ml”. Please change it to “mL”.

Reply and revision: Thank you very much for your suggestion. We have revised it in the manuscript (L161).

  1. Line 149: Please change the beginning of the sentence to “Solutions containing Zn2+were prepared…” instead of “Prepare Zn2+ solutions…”. Also change all verbs to the past tense.

Reply and revision: Thank you very much for your suggestion. We have revised it in the manuscript (L166-167).

  1. Line 149: “100 mg L-1”. In the manuscript the authors usually use mg/L but here mg L-1 is written instead. Please uniform it to always use the same style in the full manuscript. The same with all volumes mentioned since in the lines 151 and 152 “ml” is written instead of “mL”. Please check the rest of the manuscript regarding these issues.

Reply and revision: Thank you very much for your suggestion. We have revised it in the manuscript (L166). And we have unified the units in the full manuscript.

  1. Line 171: “were conduct”. Please change it to “were conducted”.

Reply and revision: Thank you very much for your suggestion. We have revised it in the manuscript (L189).

  1. Line 173: “Physico-chemical characterization of and surface properties”. Please remove the “of” word.

Reply and revision: Thank you very much for your suggestion. We have revised it in the manuscript (L191).

  1. Line 174: “element composition”. Please change it to “elemental composition”.

Reply and revision: Thank you very much for your suggestion. We have revised it in the manuscript (L192).

  1. Line 175 and 193: Please write “elements” instead of “element”. Moreover, please write the wt% of each major element after their name, otherwise the readers will need to search for your previous publication to check the ICP results.

Reply and revision: Thank you very much for your suggestion. We have revised it in the manuscript (L193, 211).

  1. Line 178: “these” instead of “this” because you are referring to the plural.

Reply and revision: Thank you very much for your suggestion. We have revised it in the manuscript (L196).

  1. Line 177: The full Fig.1 is equal to the previous study performed by the authors in Ref. [45]. The same image cannot be published two times in different papers and Journals. Since the previous study focused in Cu2+with the same adsorbent, the authors should only mention in this manuscript that the SEM, FTIR and BET characterization were previously performed in [45], without copying the same tables and figures. The values, on the other hand, can be reported in the text but always mention that they were made in [45]. If the authors would like to introduce some extra characterization in this manuscript, I would suggest they report EDS spectra and/or EDS mapping images of the biomass.

Reply and revision: Thank you very much for your suggestion. We have deleted the Figure 1 in the manuscript.

  1. The experiment was performed how many times? The graphics in the manuscript do not have error bars. Is the given data reproducible?

Reply and revision: Thank you very much for your suggestion. The experiment was performed one time, because in the previous adsorption experiment, we found that the difference between repetitions was very small. At the same time, through literature research, we found that there was no repetition in many adsorption experiments[1-5], so there was no repetition experiment in this study.

  1. Line 267: “…verified that he adsorption…”. Typo in “he”, since it should be “the”.

Reply and revision: Thank you very much for your suggestion. We have revised it in the manuscript (290).

  1. The conclusions are too general. Some quantification on the zinc abatement is required.

Reply and revision: Thank you very much for your suggestion. We have revised it in the manuscript (L338,344,345,346,347).

Reference

[1] Bourliva Anna, Michailidis Kleopas, Sikalidis Constantine, Filippidis Anestis, Betsiou Maria. Adsorption of Cd(II), Cu(II), Ni(II) and Pb(II) onto natural bentonite: study in mono- and multi-metal systems[J]. Environmental Earth Science,2015,73:5435–5444.

[2] Ting-Chu Hsu, Chung-Chin Yu, Chin-Ming Yeh. Adsorption of Cu2+ from water using raw and modified coal fly ashes[J]. Fuel,2008,87:1355–1359.

[3] Jaime Pizarro, Ximena Castillo, Sebastián Jara, Claudia Ortiz, Patricio Navarro, Héctor Cid,

Héctor Rioseco, Daniel Barros, Nelson Belzile. Adsorption of Cu2+ on coal fly ash modified with functionalized mesoporous silica[J]. Fuel, 2015,156:96–102.

[4] N.N. Nortey Yeboah, Christopher R. Shearer, Susan E. Burns, Kimberly E. Kurtis. Characterization of biomass and high carbon content coal ash for productive reuse applications[J]. Fuel,2014,116:438–447.

[5] Ibtissem Ghorbel-Abid, Malika Trabelsi-Ayadi. Competitive adsorption of heavy metals on local landfill clay[J]. Arabian Journal of Chemistry, 2015,8:25–31.

Reviewer 2 Report

Please provide reflection about zinc source and effects in the abstract.

Please correct this "2.3. Physico-chemical characterization of and surface properties".

"only guaranteed reagents were used in the experiment", not clear, maybe you want to say that it is with certain grade chemical!

Figure 1, it is entirely from reference 45! same goes to Table 1, Figure 2, and Figure 3.

Table 3 is confusing when compared to what already reported in reference 45.

Author Response

Dear Editor,

We are very grateful for you and the reviewers’ comments and suggestions about this paper. Those comments are very valuable and helpful for revising and improving our manuscript. According to these revision suggestions, we have carefully addressed the comments and made revision in detail as following, and all the modified parts in the manuscript have been marked in red.

Sincerely yours,

Lei Xu.

Issues raised by Reviewer 2

Comments and Suggestions for Authors:

  1. Please provide reflection about zinc source and effects in the abstract.

Reply and revision: Thank you very much for your suggestion. We have revised it in the manuscript (L38-40,47-49).

  1. Please correct this "2.3. Physico-chemical characterization of and surface properties".

Reply and revision: Thank you very much for your suggestion. We have revised it in the manuscript (L138).

  1. "only guaranteed reagents were used in the experiment", not clear, maybe you want to say that it is with certain grade chemical!

Reply and revision: Thank you very much for your suggestion. We have revised it in the manuscript (L151).

  1. Figure 1, it is entirely from reference 45! same goes to Table 1, Figure 2, and Figure 3.

Reply and revision: Thank you very much for your suggestion. Table 1 and Figure 1 were same to the reference 45, because the same image cannot be published two times in different papers and Journals. So we have deleted the Figure 1 in the manuscript and only mention in this manuscript that the SEM, FTIR and BET characterization were previously performed in [45], without copying the same figures.

  1. Table 3 is confusing when compared to what already reported in reference 45.

Reply and revision: Thank you very much for your suggestion. There are some differences between the data in Table 3 and the previously published papers. This may be due to the differences in the adsorption processes and mechanisms of zinc and copper between the two materials, resulting in differences in their thermodynamic characteristics.

Reviewer 3 Report

The topic is related to the journal. In order to improve the quality of this paper and meet the standard of INT J ENV RES PUB HE, major revisions need to be done as the following suggestions.

1. English check MUST be done for the whole manuscript.

2. Some keywords have little relevance to the content of the article. Authors should use more specific keywords.

3. In the section of Introduction, the innovation of the research needs to be properly addressed. The material was modified by APS and HMS matrix, but it is not clearly elaborated. Meanwhile, the section of Introduction should be compressed.

4. The biomass ash has a high silicon content itself, why further modify it with mesoporous silica?

5. Section 2.2: The modification ash method was not clearly described and details should be given on how biomass ash was modified by 10% APS and HMS matrix?

6. Section 2.4: How many parallel samples were set up in the adsorption experiment and how was the error controlled?

7. Section 2.4.2-2.4.3: The maximum adsorption efficiency of biomass ash and modified materials was found to be near pH 6. Why was pH 5 chosen for the adsorption equilibrium and kinetics experiment?

8. Section 2.4.2: What did “spectra-220” mean?

9. Section 3.1: The author mentioned that the elemental composition of biomass ash was determined by ICP-OES in this paper, why the result is not listed but quoted from an earlier article?

10. Section 3.1: The wavelengths listed in the FTIR analysis did not correspond to those in the graph.

11. Table 1: What “HMS-NH2” and “Fly ash + HMS + 10% APS” stand for respectively was not specified earlier.

12. Line 221: “When pH was >6, the adsorption of Zn2+ was weak.” Please check if the representation was accurate, as the graph showed an increasing trend in the adsorption capacity when pH > 6.

13. Section 3.3: The meaning of parameters in the formula was not listed. The author could utilize the following papers to elaborate on this discussion. (Efficient removal of hexavalent chromium through the adsorption-reduction-adsorption pathway by iron-clay biochar composite prepared from Populus nigra, Separation and Purification Technology, 2022, 285, 120386)

14. Figure 3: The correspondence between each control group and the vertical coordinate should be clearly marked.

Author Response

Dear Editor,

We are very grateful for you and the reviewers’ comments and suggestions about this paper. Those comments are very valuable and helpful for revising and improving our manuscript. According to these revision suggestions, we have carefully addressed the comments and made revision in detail as following, and all the modified parts in the manuscript have been marked in red.

Sincerely yours,

Lei Xu.

Issues raised by Reviewer 3

Comments: The topic is related to the journal. In order to improve the quality of this paper and meet the standard of INT J ENV RES PUB HE, major revisions need to be done as the following suggestions.

  1. English check MUST be done for the whole manuscript.

Reply and revision: Thank you very much for your suggestion. We have checked all the manuscript and modified all the language of the whole manuscript, and all the changes have been marked in red.

  1. Some keywords have little relevance to the content of the article. Authors should use more specific keywords.

Reply and revision: Thank you very much for your suggestion. We have changed the keywords in the manuscript (L26).

  1. In the section of Introduction, the innovation of the research needs to be properly addressed. The material was modified by APS and HMS matrix, but it is not clearly elaborated. Meanwhile, the section of Introduction should be compressed.

Reply and revision: Thank you very much for your suggestion. We have revised the introduction in the manuscript and some sentences in the original introduction have been deleted (L100-101,104-105).

  1. The biomass ash has a high silicon content itself, why further modify it with mesoporous silica?

Reply and revision: Thank you very much for your suggestion. Although the biomass ash can be used as an economic and environmental friendly adsorbent, it is reported that the adsorption capacity of natural biomass ash for specific heavy metals in aqueous solution is lower than that of some commercial or modified adsorbents. This will lead to the low efficiency of using the nature biomass ash collected from biomass power plants as an adsorbent to remove heavy metal ions from industrial wastewater. The adsorption capacity of biomass ash to heavy metals in wastewater can be improved through appropriate modification methods, so as to improve its ability to remediate heavy metal polluted wastewater. Therefore, it is necessary to explore a suitable method for modifying biomass ash.

  1. Section 2.2: The modification ash method was not clearly described and details should be given on how biomass ash was modified by 10% APS and HMS matrix?

Reply and revision: Thank you very much for your suggestion. The specific steps of the modification method have been added into the manuscript (L128-137).

  1. Section 2.4: How many parallel samples were set up in the adsorption experiment and how was the error controlled?

Reply and revision: Thank you very much for your suggestion. The experiment was performed one time, because in the previous adsorption experiment, we found that the difference between repetitions was very small. At the same time, through literature research, we found that there was no repetition in many adsorption experiments[1-5], so there was no repetition experiment in this study.

  1. Section 2.4.2-2.4.3: The maximum adsorption efficiency of biomass ash and modified materials was found to be near pH 6. Why was pH 5 chosen for the adsorption equilibrium and kinetics experiment?

Reply and revision: Thank you very much for your suggestion. In fact, at pH > 6 the weak removal of Zn2+ resulting from the precipitation of copper species such as hydroxides or carbonates must be considered, in accordance with the distribution of metal species as a function of pH. So in order to avoid the influence of other factors on adsorption, we chose pH 5 for the adsorption equilibrium and kinetics experiment.

  1. Section 2.4.2: What did “spectra-220” mean?

Reply and revision: Thank you very much for your suggestion. We have revised the expression in the manuscript (L173-174).

  1. Section 3.1: The author mentioned that the elemental composition of biomass ash was determined by ICP-OES in this paper, why the result is not listed but quoted from an earlier article?

Reply and revision: Thank you very much for your suggestion. Because the biomass ash used in this study is consistent with the biomass ash in our previous study, the previous data are quoted. In order to make the results more clearer, we add the percentage of major elements in the text (L193).

  1. Section 3.1: The wavelengths listed in the FTIR analysis did not correspond to those in the graph.

Reply and revision: Thank you very much for your suggestion. We have revised the expression in the manuscript (L202,204).

  1. Table 1: What “HMS-NH2” and “Fly ash + HMS + 10% APS” stand for respectively was not specified earlier.

Reply and revision: Thank you very much for your suggestion. We have added the explanation in the manuscript (Table 1).

  1. Line 221: “When pH was >6, the adsorption of Zn2+was weak.” Please check if the representation was accurate, as the graph showed an increasing trend in the adsorption capacity when pH > 6.

Reply and revision: Thank you very much for your suggestion. We have changed the keywords in the manuscript. In our experiment, it is true that the removal rate is the highest when the pH is around 6, but according to literature research, when pH was >6, the adsorption of Zn2+ was weak, which can be attributed to the precip-itation of Zn2+ species such as carbonates or hydroxides. So we think that the most efficient adsorption of Zn2+ on biomass ash and modified biomass ash should be pH 5-6.

  1. Section 3.3: The meaning of parameters in the formula was not listed. The author could utilize the following papers to elaborate on this discussion. (Efficient removal of hexavalent chromium through the adsorption-reduction-adsorption pathway by iron-clay biochar composite prepared from Populus nigra, Separation and Purification Technology, 2022, 285, 120386)

Reply and revision: Thank you very much for your suggestion. We have added the meaning of the parameters in the manuscript and quoted the literature you recommended (L247-252, reference 57).

  1. Figure 3: The correspondence between each control group and the vertical coordinate should be clearly marked.

Reply and revision: Thank you very much for your suggestion. We have modified the Figure 2.

Reference

[1] Bourliva Anna, Michailidis Kleopas, Sikalidis Constantine, Filippidis Anestis, Betsiou Maria. Adsorption of Cd(II), Cu(II), Ni(II) and Pb(II) onto natural bentonite: study in mono- and multi-metal systems[J]. Environmental Earth Science,2015,73:5435–5444.

[2] Ting-Chu Hsu, Chung-Chin Yu, Chin-Ming Yeh. Adsorption of Cu2+ from water using raw and modified coal fly ashes[J]. Fuel,2008,87:1355–1359.

[3] Jaime Pizarro, Ximena Castillo, Sebastián Jara, Claudia Ortiz, Patricio Navarro, Héctor Cid,

Héctor Rioseco, Daniel Barros, Nelson Belzile. Adsorption of Cu2+ on coal fly ash modified with functionalized mesoporous silica[J]. Fuel, 2015,156:96–102.

[4] N.N. Nortey Yeboah, Christopher R. Shearer, Susan E. Burns, Kimberly E. Kurtis. Characterization of biomass and high carbon content coal ash for productive reuse applications[J]. Fuel,2014,116:438–447.

[5] Ibtissem Ghorbel-Abid, Malika Trabelsi-Ayadi. Competitive adsorption of heavy metals on local landfill clay[J]. Arabian Journal of Chemistry, 2015,8:25–31.

Reviewer 4 Report

Dear Authors,

explain in the text

1.       1)Provide the formula specifying the amount of sorption (research methodology)

2.       2)Formula 4, 5, 6, 7 and 8. Explain what Ce, q etc. means.

3.       3)Figure 3 is not legible.

4.      4) % of sorption shown in the figure 3 is not discussed in the text.

5.      5) It was also not explained why the modification of the material influenced the kinetics of the process.

Author Response

Dear Editor,

We are very grateful for you and the reviewers’ comments and suggestions about this paper. Those comments are very valuable and helpful for revising and improving our manuscript. According to these revision suggestions, we have carefully addressed the comments and made revision in detail as following, and all the modified parts in the manuscript have been marked in red.

Sincerely yours,

Lei Xu.

Issues raised by Reviewer 4

Comments:

1.Provide the formula specifying the amount of sorption (research methodology).

Reply and revision: Thank you very much for your suggestion. We have added the formula in the manuscript (L151-157).

  1. Formula 4, 5, 6, 7 and 8. Explain what Ce, q etc. means.

Reply and revision: Thank you very much for your suggestion. We have added the explain of the meaning of codes in formula (L247-252, 277-278).

  1. Figure 3 is not legible.

Reply and revision: Thank you very much for your suggestion. We have modified Figure 2 in the manuscript.

  1. % of sorption shown in the figure 3 is not discussed in the text.

Reply and revision: Thank you very much for your suggestion. We have added the discussion in the manuscript (L300-301).

  1. It was also not explained why the modification of the material influenced the kinetics of the process.

Reply and revision: Thank you very much for your suggestion. The possible causes of the change of the kinetics of the process were added in the manuscript (L324-326).

Round 2

Reviewer 1 Report

The authors revised the manuscript according to the suggestions. However, I have now one very serious concern: all the improved adsorption capacities now reported in the final version of the manuscript include values above 100%! Even one value was reported as 1145%! For example: "Cu2+, Pb2+, Cr3+ were improved by 410%-524%, 340%-623% and 231%-1145%" (L102); "the specific surface area of 331 the modified materials has been greatly improved by 769%" (L332, Conclusions). This is impossible! It simply cannot be and it cannot be published like this! 

Please calculate these values properly. The maximum of a value cannot be higher than 100% in any circumstance. In fact, the authors reported Zn2+ removal in % in Fig. 2 and all values are below 100% which it is normal, so just calculate the improved adsorption % in the same way as a difference of values in % as you calculated for Fig. 2. However, if the authors cannot calculate it, it is better to simply remove these higher than 100% values.

Author Response

Dear Editor,

We are very grateful for you and the reviewers’ comments and suggestions about this paper. Those comments are very valuable and helpful for revising and improving our manuscript. According to these revision suggestions, we have carefully addressed the comments and made revision in detail as following, and all the modified parts in the manuscript have been marked in red.

Sincerely yours,

Lei Xu.

Issues raised by Reviewer 1

Comments: The authors revised the manuscript according to the suggestions. However, I have now one very serious concern: all the improved adsorption capacities now reported in the final version of the manuscript include values above 100%! Even one value was reported as 1145%! For example: "Cu2+, Pb2+, Cr3+ were improved by 410%-524%, 340%-623% and 231%-1145%" (L102); "the specific surface area of 331 the modified materials has been greatly improved by 769%" (L332, Conclusions). This is impossible! It simply cannot be and it cannot be published like this!

Please calculate these values properly. The maximum of a value cannot be higher than 100% in any circumstance. In fact, the authors reported Zn2+ removal in % in Fig. 2 and all values are below 100% which it is normal, so just calculate the improved adsorption % in the same way as a difference of values in % as you calculated for Fig. 2. However, if the authors cannot calculate it, it is better to simply remove these higher than 100% values.

Reply and revision: Thank you very much for your suggestion. (1) In the manuscript, the expression is “the adsorption capacity of Cu2+, Pb2+, Cr3+ were improved by 410%-524%, 340%-623% and 231%-1145%", this refers to the adsorption capacity (mg/g), according to your suggestion, we feel that this expression is not standardized and should be consistent with the expression in Figure 2 in the manuscript (removal rate %), so we have modified it in the manuscript (L103-105). (2) In this study, the specific surface area of modified biomass ash is 9 times that of the material before modification but removal rate of Zn2+ in solution is just 21.6% higher than that of biomass ash, and we have modified this statement in the manuscript according to your suggestion (L338-339).

Reviewer 3 Report

From my point of veiw, the revised manusript can be accepted for the publication after several minor modifications as below.

Lines 152&244&273, "Where" should be "where".

Line 320, "NH2" needs a subscript of "2".

Line 341, "times of the" needs improving.

Author Response

Dear Editor,

We are very grateful for you and the reviewers’ comments and suggestions about this paper. Those comments are very valuable and helpful for revising and improving our manuscript. According to these revision suggestions, we have carefully addressed the comments and made revision in detail as following, and all the modified parts in the manuscript have been marked in red.

Sincerely yours,

Lei Xu.

Issues raised by Reviewer 3

Comments: From my point of view, the revised manuscript can be accepted for the publication after several minor modifications as below.

  1. Lines 152&244&273, "Where" should be "where".

Reply and revision: Thank you very much for your suggestion. We have revised it in the manuscript (L156,248,278).

  1. Line 320, "NH2" needs a subscript of "2".

Reply and revision: Thank you very much for your suggestion. We have revised it in the manuscript (L326).

  1. Line 341, "times of the" needs improving.

Reply and revision: Thank you very much for your suggestion. We have revised it in the manuscript (L348-349).

This manuscript is a resubmission of an earlier submission. The following is a list of the peer review reports and author responses from that submission.